# The Healthy Eating Assessment Tool (HEAT): A Simplified 10-Point Assessment of CHILD-2 Dietary Compliance for Children and Adolescents with Dyslipidemia

**DOI:** 10.3390/nu15041062

**Published:** 2023-02-20

**Authors:** Sara DiLauro, Jonathan P. Wong, Tanveer Collins, Nita Chahal, Brian W. McCrindle

**Affiliations:** 1Labatt Family Heart Centre, The Hospital for Sick Children, Toronto, ON M5G 1X8, Canada; 2Peter Munk Cardiac Centre, University Health Network, Toronto, ON M5G 2N2, Canada; 3Department of Pediatrics, University of Toronto, Toronto, ON M5S, Canada; 4Department of Nutritional Sciences, University of Toronto, Toronto, ON M5S, Canada

**Keywords:** dyslipidemia, diet, obesity, nutritional status

## Abstract

Traditional dietary assessment tools used to determine achievement of cholesterol-lowering dietary targets, defined in the Cardiovascular Health Integrated Lifestyle Diet (CHILD-2), are time intensive. We sought to determine the utility of the Healthy Eating Assessment Tool (HEAT), a simplified 10-point dietary assessment tool, in relation to meeting dietary cut points of the CHILD-2, as well as its association with markers of adiposity and lipid variables. We performed a 2-year single-center, prospective cross-sectional study of pediatric patients with dyslipidemia. HEAT score associations with meeting CHILD-2 fat targets were modest. Only patients with the highest HEAT scores (good 43%, excellent 64%) met the CHILD-2 cut point of <25% total fat calories (*p* = 0.03), with a non-significant trend for limiting the percentage of daily saturated fat to <8% (excellent 64%), and no association with cholesterol intake. There were more consistent associations with markers of adiposity (body mass index z-score r = −0.31, *p* = <0.01 and waist-to-height ratio r = −0.31, *p* = <0.01), and there was no independent association with lipid levels. While fat-restricted diets are safe, they are not particularly effective for treatment of dyslipidemia or for weight management alone. The HEAT may be a more useful and simplified way of assessing and tracking broader dietary goals in clinical practice.

## 1. Introduction

Dietary therapy, along with physical activity prescription, are the key lifestyle modifications and first line of therapy for the treatment of dyslipidemia. Lipid-lowering dietary therapy is considered the cornerstone of treatment because it is a safe and effective way to lower lipid levels in children and adolescents [1]. A low-fat, low-sugar, and high-fiber diet that relies primarily on vegetables and fruits, whole grains, lean meats, fish, lentils and legumes, and low- and non-fat dairy products is recommended. The two-stage dietary approach of the Cardiovascular Health Integrated Lifestyle Diet (CHILD), composed of the CHILD-1 and CHILD-2, provides dietary goals for children and adolescents, aged 2–21 years, with dyslipidemia [1]. The CHILD-1 is the first stage in dietary change and recommends total fat intake be limited to 25–30% of daily calories. Of the total fat intake, saturated fat is to be limited to 8–10% of daily kcal/EER, while mono and polyunsaturated fat consumption is encouraged up to 20% of daily kcal/EER. Trans fat is to be avoided as much as possible, and dietary cholesterol is to be limited to <300 mg/day. The remaining 70% of dietary energy should be composed of 15–20% proteins and 50–55% carbohydrates, with a focus on fiber-rich foods such as vegetables, fruit, and whole grains. If dyslipidemia persists after 3 months, the CHILD-2 is recommended, which suggests a further restriction of saturated fat to <7% of daily kcal/EER, monounsaturated fat to 10% of daily kcal/EER, and limitation of dietary cholesterol to <200 mg/day (Table 1). A referral to a registered dietitian (RD) for family medical nutrition therapy is strongly recommended.

Traditional dietary assessment tools, such as 24 h dietary recalls, food records, or food frequency questionnaires, are unable to provide rapid and accurate assessment of dietary fat, cholesterol, or fiber intake for timely assessment and nutrition care plan formulation during routine clinical visits. In addition, they do not incorporate eating behaviors such as skipping meals. There is a need for a dietary assessment tool that can quickly determine adherence to cholesterol-lowering diets as outlined by the CHILD-2. The simplified 10-point Healthy Eating Assessment Tool (HEAT) was created by the RDs in the Lipid Clinic at The Hospital for Sick Children as a means to assess a patient’s overall dietary quality and behaviors during time-limited individual clinic counselling sessions.

The aims of this study were (1) to determine the association between HEAT scores and components of the CHILD-2, (2) to determine the association between HEAT scores and markers of adiposity, and (3) to determine the association between HEAT scores and lipid biomarkers.

## 2. Materials and Methods

### 2.1. Study Design and Population

This was a single-center, prospective cross-sectional study conducted at The Hospital for Sick Children in Toronto, Canada over a 2-year period. All of the patients that were monitored in the Lipid Clinic for dyslipidemia were screened for eligibility. Inclusion criteria were those from ages 2 to 18 years who had completed food records, were assessed by an RD, and were assigned a HEAT score during any follow-up visit. Exclusion criteria were those who had only attended an initial Lipid Clinic visit, had not met with the RD in the past 2 years, or had never completed food records for any of their previous appointments. Those who had a diagnosis that significantly impacted dietary intake or those following a ketogenic diet were also excluded. Eligible participants were approached, and written informed consent was obtained by the study team. This study received institutional ethics approval by The Hospital for Sick Children’s Research Ethics Board.

### 2.2. Healthy Eating Assessment Tool (HEAT)

The HEAT is based on a RD-led interview that uses a 10-point assessment tool to assess a patient’s overall dietary quality during the Lipid Clinic’s time-limited counselling sessions (Figure 1 and Appendix A—Lipid Clinic Assessment Form—HEAT tool component highlighted in yellow). During the RD-led interview, patients/caregivers are asked specific questions on 10 different subcategories of their diet, as shown in Figure 1. Each diet subcategory is given either a score of 1 point, 0.5 points, or 0 points depending on the patient’s consumption habits and the quality of the foods consumed, based on dietary recommendations for dyslipidemia and healthy eating [2,3,4]. The HEAT score is tallied through addition of the points awarded for each of the 10 subcategory scores. Appendix B provides a detailed description of how points are assessed and awarded for each diet subcategory. A total score of 0–4.5 out of 10 indicates a poor score, 5–6.5 fair, 7–8.5 good, and 9–10 indicates an excellent score.

### 2.3. Data Collection

Collected data included age, sex, anthropometrics (weight, height, body mass index (BMI), and waist circumference), fasting serum lipid levels (total cholesterol, LDL-C, HDL-C, and triglycerides), fasting serum glucose, blood pressure, medications, nutritional supplements, physical activity and screen time (hours per week), and dietary intake. To assess dietary intake for this study, patients/caregivers completed 7-day food records prior to their clinic visit. These 7-day food records were used instead of the 4-day food records routinely used in the Lipid Clinic as some food items on the HEAT are assessed by their weekly consumption rather than their daily consumption (e.g., egg yolks, restaurant/fast-food) and a 7-day timeframe reflects a more comprehensive dietary intake. The 7-day food records were reviewed and verified in detail by the RD/nutrition research student during the clinic study visit. If portion sizes or food items on the food records were unclear, patients/caregivers were asked to clarify the amount consumed using food models and visual images to help estimate accurate portion sizes.

### 2.4. Dietary Analysis

Content from the 7-day food records was inputted electronically into the nutrition analysis software program, ESHA Research Food Processor SQL (version 10.12.2), by the RD/nutrition research student for analysis of the macronutrient content (i.e., total calories, total fat, saturated fat, trans fat, unsaturated fat, omega-3 and -6, cholesterol, total carbohydrates, sugar, fiber (soluble and insoluble), and protein), as well as for the micronutrient content (vitamins and minerals). The 7-day food records were also analyzed to determine consumption from the four food groups on Canada’s Food Guide to Healthy Eating (vegetables and fruit, grain products, dairy and alternatives, and meat and alternatives) [5]. All relevant data extracted from the ESHA nutrient data reports were inputted into the secure, web-based data management program, Research Electronic Data Capture (REDCap), prior to analysis.

### 2.5. Outcomes

The primary outcome of this study was the association between HEAT scores and dietary component targets of the CHILD-2. The secondary outcomes of this study were the association between HEAT scores and measures of adiposity, as well as the association between HEAT scores and lipid biomarkers.

### 2.6. Statistical Analysis

Data analysis was performed using SAS software v9.3 (The SAS Institute, Cary, NC, USA). Descriptive data were expressed as means (standard deviation) and frequencies according to the level of measurement of the variable. To determine the association of HEAT score as a continuous variable with average daily nutrient breakdown from food record analysis, Pearson’s correlation coefficients were derived. To determine the association of HEAT score categories and the achievement of CHILD-2 dietary targets, the Mantel–Haenszel chi-squared test for trends was used. To determine the association between HEAT score categories and markers of adiposity, ANOVA was used. To determine the association of HEAT score as a continuous variable with lipid profile variables, generalized linear regression models incorporating relevant covariates were used. *p*-values < 0.05 were considered statistically significant.

## 3. Results

### 3.1. Study Population

Characteristics of the 70 enrolled participants are shown in Table 2. Participants were recruited across the whole spectrum of HEAT scores. The underlying dyslipidemia was heterogeneous across the study population, and as some were treated with lipid-lowering medication, the lipid values were likewise heterogeneous. The underlying dyslipidemia was heterozygous familial hypercholesterolemia for 33 (47%) patients (19 treated with a statin), combined dyslipidemia for 16 (23%) patients (2 treated with medication), mild dyslipidemia with elevated LDL-C for 14 (20%) patients, isolated low HDL-C for 4 (13%) patients, and elevated lipoprotein(a) for 3 (10%) patients. All patients were recommended for dietary management aimed at achieving the CHILD-2 dietary targets.

### 3.2. HEAT Score and Components of CHILD-2

Correlations between HEAT scores and CHILD-2 dietary components as continuous variables are listed in Table 3. A higher HEAT score was significantly associated with a lower total fat percent of total daily calories, and with a higher dietary fiber and fruit and vegetable intake. There was no significant correlation between HEAT score and the remaining dietary intake variables or the weekly total of moderate-to-vigorous physical activity.

The association between HEAT score categories and the achievement of CHILD-2 recommendations is shown in Figure 2. The association of HEAT score category with the proportion meeting the recommendation to limit total fat to <25% of total daily calories was significant but non-linear, with the greatest proportions meeting the cut point only with the highest HEAT score categories (“good” and “excellent”). A similar non-linear trend was noted for the proportion meeting the recommendation to limit saturated fat to <8% of total daily calories (“excellent”), although it was not statistically significant (*p* = 0.08). There was no association of the proportion meeting the recommendation to limit total cholesterol to <200 mg per day with HEAT score category.

### 3.3. HEAT Score and Markers of Adiposity

The association between HEAT score category and markers of adiposity is shown in Figure 3. The HEAT score categories of “poor” and “fair” were significantly associated with both higher BMI z-scores and higher waist-to-height ratios. Using the HEAT score as a continuous variable, there was a significant negative correlation between HEAT score and BMI z-score (r = −0.31, *p* = <0.01), as well as a significant negative correlation between HEAT score and waist-to-height ratio (r = −0.31, *p* = <0.01).

### 3.4. HEAT Score and Lipid Variables

In generalized linear regression models adjusted for age, sex, hours of moderate-to-vigorous physical activity, hours of screen time, lipid-lowering medication, BMI z-score, and waist-to-height ratio, no lipid variable was significantly associated with HEAT score (LDL-C *p* = 0.42, R^2^ = 0.33; HDL-C *p* = 0.22, R^2^ = 0.36; triglycerides *p* = 0.20, R^2^ = 0.28; non-HDL-C *p* = 0.37, R^2^ = 0.33). Details of the full regression models are provided in Table 4.

## 4. Discussion

In this prospective cross-sectional study of pediatric patients with dyslipidemia, we noted that the HEAT score was modestly associated with CHILD-2 daily fat percentage targets and was more consistently associated with the markers of adiposity, specifically BMI z-score and waist-to-height ratio. These results suggest that the HEAT may be a pragmatic tool to assess and track broader dietary goals in children, particularly useful for identifying children with suboptimal dietary patterns that may put them at risk of obesity and dyslipidemia and who might benefit from targeted dietary counselling.

Atherosclerotic cardiovascular disease (ASCVD) is the predominant cause of death in developed countries [2]. Atherosclerosis starts in childhood and is associated with risk factors, such as obesity and dyslipidemia, that continue into adulthood [3,6]. Conventional cardiovascular risk factors, such as obesity and dyslipidemia, have been linked to the development of early markers of atherosclerosis, specifically fatty streaks and fibrous plaques, in children [7]. Childhood obesity, defined as a BMI percentile greater than the 95th percentile, is a global health crisis that has been steadily worsening over the past decades with large population-based studies showing a tenfold increase in the past four decades [4]. Lifestyle interventions are the cornerstone of the treatment and prevention of childhood obesity, and our study aimed to assess the HEAT as a time-efficient tool to assess dietary compliance with a low-fat diet alongside achievement of broader healthy eating dietary goals.

Excessive intake of dietary fats, specifically in the form of saturated fat, trans fat, and cholesterol, is a contributor to dyslipidemia and obesity. Children over the age of 12 months can safely follow a fat-restricted diet under the supervision of an RD [1], although the long-term efficacy and safety of a dietary intervention beginning at 7 months of age were noted, along with a number of favorable outcomes [8,9,10]. Dietary intervention studies have shown the safety and efficacy of low-fat and low-cholesterol diets in children [11,12]. The Dietary Intervention Study in Children (DISC) was a randomized control trial that studied American children aged 8–11 years (n = 663) with modest LDL-C elevation by comparing a 3-year intensive dietary intervention that limited dietary fat (28% total fat of total daily calories and 10% saturated fat of total daily calories) and cholesterol (95 mg per day) with the standard of care [13,14]. There were greater reductions in dietary total fat, saturated fat, and cholesterol in the intervention group, as well as a statistically significant adjusted relative reduction in LDL-C of 0.08 mmol/L (*p* = 0.02). There were no differences between groups in height, serum ferritin, sexual maturation, or BMI. Based on the small effect size, one might question the efficacy of such a diet in reducing LDL-C levels, particularly since both groups showed similar increases over time in their BMIs. Similar results were found in the Special Turku Coronary Risk Factor Intervention Project for Babies (STRIP) randomized control trial that followed children from age 7 months to adulthood (n = 1062), with an intervention group that received dietary counselling biannually until 7 years and annually until 20 years, aiming to achieve a low-fat (<30–35% total fat of total daily calories and unsaturated-to-saturated-fat ratio of 2:1) and low-cholesterol (<200 mg per day) diet [8]. They achieved a reduced dietary intake of saturated fat and an increased intake of polyunsaturated fat when comparing the intervention and control groups. Additionally, the dietary intervention group had greater baseline-adjusted reductions in HDL-C, non-HDL-C, and apolipoproteins A1 and B, although the effect size for non-HDL-C was modest. The cohort continues to be followed now up to 20 years with favorable results [10]. Given the modest effect size on LDL-C and non-HDL-C levels, together with reductions in HDL-C, the utility of dietary intervention for pediatric dyslipidemia management, while safe, was suggested to be limited.

The limits to dietary assessment and counselling are the time intensiveness, particularly in the setting of healthcare systems with limited resources for prolonged dietary encounters, and sometimes even the availability of a qualified RD. We identified a need for a time-efficient and accurate tool to assess dietary compliance with the CHILD-2 and achievement of broader nutritional goals that could quickly identify targets for dietary intervention and could be used to monitor a patient’s progress over time. We found that a higher HEAT score was weakly correlated with a lower total fat percentage of total daily calories, as well as a higher-fiber fruit and vegetable intake. However, there was no significant association with intake of saturated fat, trans-fats, and cholesterol. These results suggest that the HEAT tool may have limited utility in the assessment of compliance with and attainment of the specific targets of the CHILD-2 diet. It is also suggested that dietary adequacy was only achieved when patients achieved the highest HEAT scores. This was true for the association with markers of adiposity. The association of HEAT score with waist-to-height ratio and BMI z-score could be harnessed to identify patients with childhood obesity that may benefit from potential dietary targets. Although the use of the HEAT tool in specialty pediatric lipid clinics may be limited, its use in the primary care setting, where there is a need to efficiently assess diet in a time-constrained encounter, may offer a potential future area of research.

A limitation of this study is the reliance on a patient and/or their caregiver’s dietary recall to inform the HEAT score. Although this is paired with other methods of analyzing dietary intake, such as food records, food frequency questionnaires, and 24 h dietary recall, there is an inherent recall bias with a dietary assessment tool administered by a healthcare professional during a clinic visit, especially given a patient/caregiver’s desire to improve the diet, which may lead them to falsely recall a better version of the diet than is correct. Additionally, the study took place in a single specialty lipid clinic, so generalizability to other lipid clinics and to primary care practices is unknown. We were also unable to assess potential intra- and inter-observer variability. A further limitation is the cross-sectional nature of this study. Future research is needed to address whether changes in HEAT score are associated with changes in lipid values and measures of adiposity.

## 5. Conclusions

Given that the HEAT tool assesses broad dietary goals, associations with nutrient breakdown and achievement of fat and cholesterol intake targets were modest, with more consistent associations with markers of adiposity. There was no independent association with lipid levels, given the underlying heterogeneity of the pediatric population with both untreated and treated dyslipidemia. While fat-restricted diets are safe, they have not been particularly effective for treatment of pediatric dyslipidemia or for weight management alone. Our results suggest that the HEAT tool may have limited utility in the assessment of compliance with and attainment of the specific targets of the CHILD-2 diet. Instead, the HEAT tool may offer a pragmatic and simplified way of assessing and tracking broader dietary goals in clinical practice.

## Figures and Tables

**Figure 1 nutrients-15-01062-f001:**
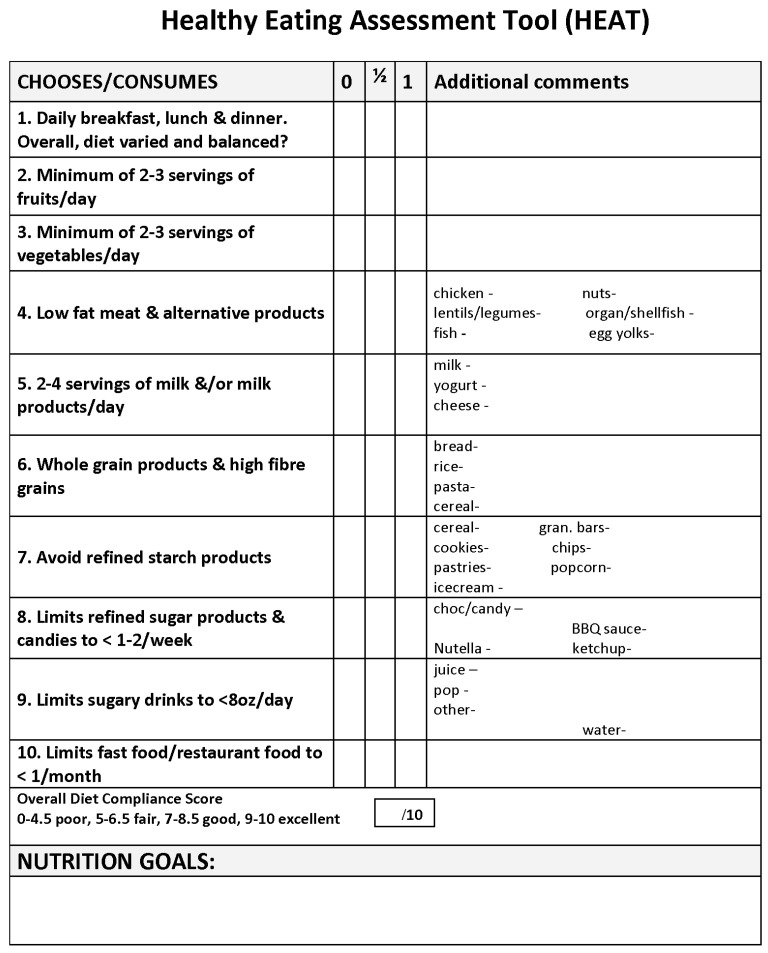
The Healthy Eating Assessment Tool (HEAT).

**Figure 2 nutrients-15-01062-f002:**
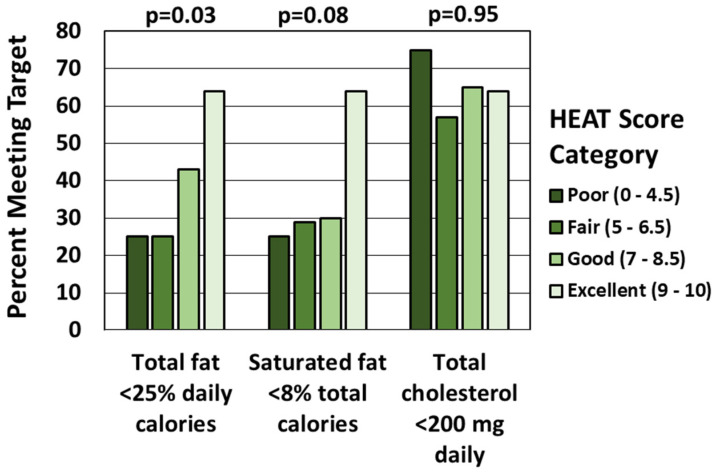
Percent of Patients Within Each HEAT Score Category Who Meet CHILD-2 Targets for Fat and Cholesterol Intake.

**Figure 3 nutrients-15-01062-f003:**
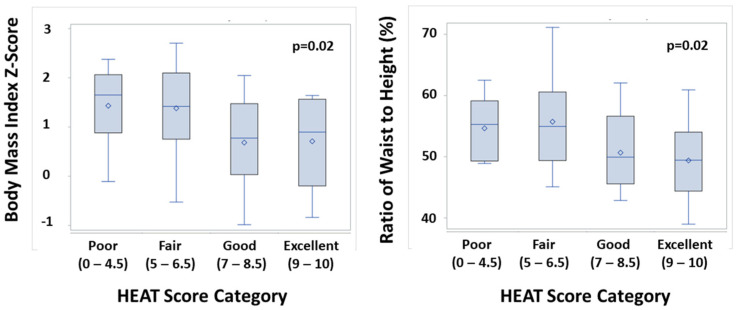
Association of HEAT Score Category with Markers of Adiposity. Box plots with boxes enclose the 25th–75th percentiles, with the line indicating the median and the diamond indicating the mean values, and the whiskers indicating the minimum and maximum values.

**Table 1 nutrients-15-01062-t001:** Dietary Components of the Cardiovascular Health Integrated Lifestyle Diet (CHILD)-2.

Dietary Components of the Cardiovascular Health Integrated Lifestyle Diet (CHILD)-2
(1) Total fat 25–30% of daily kcal intake
(2) Saturated fat ≤ 7% daily kcal intake
(3) Avoid trans fat
(4) Monounsaturated fat ~10% daily kcal intake
(5) Cholesterol < 200 mg/day (6) Reduce sugar ^1^ (7) Replace simple carbohydrates with complex carbohydrates ^1^ (8) Avoid sugar-sweetened beverages ^1^ (9) Increase dietary fish to increase omega-3 fatty acid intake ^1^

^1^ Specifically for triglyceride lowering. Adapted from [1].

**Table 2 nutrients-15-01062-t002:** Baseline Characteristics (n = 70 participants).

	Mean (±SD) or Total (%)
Age (years)	12.6 (±3.8)
Female gender	29 (41%)
HEAT score	6.7 (±2.0)
Poor (score 0–4.5)	8 (11%)
Fair (score 5–6.5)	28 (40%)
Good (score 7–8.5)	23 (33%)
Excellent (score 9–10)	11 (16%)
BMI (kg/m^2^)	23.5 (±5.9)
BMI z-score	+1.05 (±0.92)
BMI percentile (%)	78 (±24)
Waist-to-height ratio (%)	53 (±7)
Fasting blood glucose (mmol/L)	5.1 (±1.9)
Fasting lipid biomarkers	
Total cholesterol (mmol/L)	5.08 (±1.26)
HDL-C (mmol/L)	1.21 (±0.30)
LDL-C (mmol/L)	3.29 (±1.22)
Triglycerides (mmol/L) Non-HDL-C (mmol/L)	1.34 (±0.94) 3.87 (±1.22)

BMI, body mass index; HDL-C, high-density lipoprotein cholesterol; LDL-C, low-density lipoprotein cholesterol.

**Table 3 nutrients-15-01062-t003:** Association Between HEAT Score and CHILD-2 Average Daily Dietary Intake and Physical Activity.

Average Components of CHILD-2	Mean (SD)	Pearson’s Correlation with HEAT Score (r)	*p*-Value
Total fat percent of total daily calories (%)	29.2 (±11.8)	−0.27	0.02
Saturated fat percent of total daily calories (%)	9.9 (±5.0)	−0.17	0.14
Trans fat percent of total daily calories (%)	1.0 (±1.4)	−0.09	0.44
Total dietary cholesterol (mg)	218 (±182)	−0.03	0.78
Dietary fiber intake			
Average daily dietary fiber (g)	18.6 (±9.3)	0.46	<0.01
Average daily vegetables intake (servings)	1.8 (±2.4)	0.32	<0.01
Average daily fruit intake (servings)	1.7 (±1.8)	0.31	<0.01
Mono and polyunsaturated fats			
Monounsaturated fat percent of total daily calories (%)	6.9 (±4.6)	−0.18	0.13
Polyunsaturated fat percent of total daily calories (%)	3.3 (±2.2)	−0.19	0.10
Weekly total of moderate-to-vigorous physical activity (hours)	8.2 (±5.4)	0.01	0.90

**Table 4 nutrients-15-01062-t004:** Covariate-Adjusted General Linear Models for Association of Lipid Variables with HEAT Score.

**Low Density Lipoprotein-Cholesterol (LDL-C, mmol/L; model R^2^ = 0.33):**		
	**PE (SE)**	***p* value**
Intercept	7.14 (2.26)	
**HEAT score**	**−0.065 (0.078)**	**0.42**
Age (years)	−0.200 (0.043)	<0.001
Females	0.268 (0.302)	0.38
Body mass index Z-score	0.279 (0.339)	0.42
Waist to height ratio as percent	−0.023 (0.044)	0.61
Weekly moderate-vigorous physical activity (hours)	−0.015 (0.026)	0.58
Weekly screen time (hours)	0.009 (0.011)	0.42
Taking lipid lowering medication	−0.286 (0.284)	0.32

**High Density Lipoprotein-Cholesterol (HDL-C, mmol/L; model R^2^ = 0.36):**		
	**PE (SE)**	***p* value**
Intercept	1.930 (0.552)	
**HEAT score**	**−0.023 (0.018)**	**0.22**
Age (years)	−0.012 (0.010)	0.27
Females	−0.027 (0.073)	0.72
Body mass index Z-score	−0.164 (0.083)	0.06
Waist to height ratio as percent	−0.002 (0.011)	0.84
Weekly moderate-vigorous physical activity (hours)	−0.005 (0.006)	0.41
Weekly screen time (hours)	0.001 (0.003)	0.69
Taking lipid lowering medication	−0.177 (0.069)	0.02

**Non-High Density Lipoprotein-Cholesterol (Non-HDL-C, mmol/L; model R^2^ = 0.33):**		
	**PE (SE)**	***p* value**
Intercept	6.491 (2.271)	
**HEAT score**	**−0.069 (0.075)**	**0.37**
Age (years)	−0.199 (0.043)	<0.001
Females	0.166 (0.302)	0.59
Body mass index Z-score	0.246 (0.342)	0.48
Waist to height ratio as percent	0.000 (0.045)	0.99
Weekly moderate-vigorous physical activity (hours)	−0.026 (0.026)	0.32
Weekly screen time (hours)	0.016 (0.011)	0.15
Taking lipid lowering medication	−0.228 (0.283)	0.43

**Triglycerides (mmol/L; model R^2^ = 0.28):**		
	**PE (SE)**	***p* value**
Intercept	−0.242 (1.801)	
**HEAT score**	**−0.078 (0.060)**	**0.20**
Age (years)	−0.006 (0.034)	0.86
Females	0.026 (0.240)	0.92
Body mass index Z-score	−0.057 (0.271)	0.84
Waist to height ratio as percent	0.037 (0.035)	0.31
Weekly moderate-vigorous physical activity (hours)	−0.021 (0.020)	0.30
Weekly screen time (hours)	0.024 (0.009)	0.01
Taking lipid lowering medication	−0.049 (0.225)	0.83

Bold highlights adjusted association of HEAT score with individual lipid variables; PE, paramater estimate; SE, standard error.

## Data Availability

Data presented in the manuscript, code book, and/or analytic code will be made available on request.

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
