# Peer review of "The Healthy Eating Assessment Tool (HEAT): A Simplified 10-Point Assessment of CHILD-2 Dietary Compliance for Children and Adolescents with Dyslipidemia"

_nutrients, 2023, doi:10.3390/nu15041062_

Round 1

Reviewer 1 Report

In this manuscript, Di Laura and Associates evaluate how the 10-point Healthy Eating Assessment Tool (HEAT) questionnaire predicts adherence to CHILD-2-recommended cholesterol-lowering diets in dyslipidemic children. Main evaluation variable of their study were 7-day food records prepared by patients/caregivers analyzed as a continuous and graded ordinal variable. I have a series of point-to-point criticisms itemized below:

1. The Authors do not provide details about the clinical profile of the dyslipidemic children and adolescents recruited for the study. As a matter of fact, LDL-C  and triglycerides averaged 3.29 ±1.22 and  1.34±0.94 mmol/L (by the way, what about skewness of these parameters?) respectively. To me, this implies that a considerable portion (say a third or so?) of the patients included in the sample were not dyslipidemic (see the table extracted from recent Canadian Guidelines on the matter (Khoury M et al, Can J Cardiol. 2022 Aug;38(8):1168-1179. doi: 10.1016/j.cjca.2022.05.002. PMID: 35961755). Please discuss this point

2. As all research tools, questionnaires need validation. What the within- and between-observer variability of HEAT score was like (which I would'nt be surprised to be relevant)? Please discuss this point

3. As a follow-up to this point, the Authors rightly caution the reader about possible biases of food records prepared by patients/providers. Question for the Authors: Are results built upon not totally reliable parameters valid to reach solid conclusions? Please discuss this point

4. Looking for a correlation between HEAT score categories and anthropometric parameters is a useless endeavour given that BMI and waist-hip ratios are much more easy to measure than to estimate through a questionnaire. Please discuss this point.

5.If the purpose of HEAT is to check the clinical course of dyslipidemia, the missing association with lipid levels makes this clinical tool barely useful. Please discuss this point.

6. The association which emerges between HEAT scores and total fat calories<25% as well as dietary fiber intake may be statistically significant but still seems to me clinically irrelevant (See table 4 and figure 1). Please discuss this point.

Overall, it seems to me unjustified to conclude that "... These results show promise that the HEAT may be a helpful tool to assess and track broader dietary goals in children, particularly identifying children with suboptimal dietary patterns that may put them at risk of obesity and dyslipidemia and who might benefit from targeted dietary counselling...".

Author Response

In this manuscript, Di Laura and Associates evaluate how the 10-point Healthy Eating Assessment Tool (HEAT) questionnaire predicts adherence to CHILD-2-recommended cholesterol-lowering diets in dyslipidemic children. Main evaluation variable of their study were 7-day food records prepared by patients/caregivers analyzed as a continuous and graded ordinal variable. I have a series of point-to-point criticisms itemized below:

1. The Authors do not provide details about the clinical profile of the dyslipidemic children and adolescents recruited for the study. As a matter of fact, LDL-C  and triglycerides averaged 3.29 ±1.22 and  1.34±0.94 mmol/L (by the way, what about skewness of these parameters?) respectively. To me, this implies that a considerable portion (say a third or so?) of the patients included in the sample were not dyslipidemic (see the table extracted from recent Canadian Guidelines on the matter (Khoury M et al, Can J Cardiol. 2022 Aug;38(8):1168-1179. doi: 10.1016/j.cjca.2022.05.002. PMID: 35961755). Please discuss this point.

The distribution of the lipid variables was not particularly skewed, despite the heterogen. In addition, we have now added the underlying dyslipidemia diagnosis for the patients to the Results section of the manuscript. Of note, more than half of the patients with familial hypercholesterolemia were being effectively treated with a statin and had normal or borderline LDL levels. In addition, the patients with isolated low HDL and elevated lipoprotein(a) also had normal LDL and triglyceride levels. Thus, the study population was heterogeneous regarding their lipid abnormalities, but all patients required the CHILD-2 recommended diet as part of their lifestyle management.

2. As all research tools, questionnaires need validation. What the within- and between-observer variability of HEAT score was like (which I would'nt be surprised to be relevant)? Please discuss this point.

This is not a questionnaire, but a tool to facilitate dietary assessment and tracking of compliance. We agree that it would be ideal to assess intra- and inter-observer variability, but we only had one RD working in our lipid program. In addition, the interview format with the patient and family and then subsequent scoring with the HEAT tool does not lend itself to a repeated measure and assessment of intra-observer variability. We now acknowledge this as a potential Limitation in the limitation section of the Discussion.

3. As a follow-up to this point, the Authors rightly caution the reader about possible biases of food records prepared by patients/providers. Question for the Authors: Are results built upon not totally reliable parameters valid to reach solid conclusions? Please discuss this point.

The reality of a clinical scenario is that these food records are what are used in clinical practice, but together with review and discussion by the RD with the patient and family to verify and clarify what is documented. In addition, in clinical practice is very unlikely that the food records would be used to determine nutrient breakdown, as this is very resource intensive. The HEAT is much more than the food records alone, and we believe the HEAT offers a pragmatic approach to the assessment, scoring and tracking of achievement of dietary goals, as well as a structure for counseling.

4. Looking for a correlation between HEAT score categories and anthropometric parameters is a useless endeavour given that BMI and waist-hip ratios are much more easy to measure than to estimate through a questionnaire. Please discuss this point.

We respectfully disagree, and we are not sure if we understand the point completely. First, we assessed waist:height ratio, not waist:hip. Regardless of the underlying dyslipidemia, increased adiposity would be expected to have a negative impact on the lipid profile. The HEAT score is a global assessment of dietary quality and intake, more so than just the output of the nutrient analysis of food records. Hence, we feel that determination of the association of the HEAT score with markers of adiposity is important, and we do show that greater compliance with broader dietary recommendations reflected in a high HEAT score is associated with a lower level of adiposity, albeit in this cross-sectional study. As noted in our discussion we feel that the role of dietary management of dyslipidemia may be less in trying to reduce dyslipidemia and more to prevent or manage increased adiposity. We therefore feel that this aspect of the analysis is justified.

5.If the purpose of HEAT is to check the clinical course of dyslipidemia, the missing association with lipid levels makes this clinical tool barely useful. Please discuss this point.

The purpose of the HEAT tool is to provide a pragmatic assessment and scoring of compliance with global healthy dietary recommendations, without having to perform a resource intensive analysis of nutrient composition in order to determine achievement of targets of the CHILD-2, which may of limited relevance. The tool is not meant to “check the clinical course of dyslipidemia” which, with the exception of combined dyslipidemia and hypertriglyceridemia, would be expected to have a minimal impact on lipid values. The lack of an association with lipid values is not a fault of the HEAT for this cross-sectional study, but a reflection of the underlying heterogeneity of the study population. What is needed is a longitudinal study whereby changes in HEAT may have a greater, but probably still limited, association with changes in lipid values. This is a next step. This has been added at the end of the Limitations section of the Discussion.

6. The association which emerges between HEAT scores and total fat calories<25% as well as dietary fiber intake may be statistically significant but still seems to me clinically irrelevant (See table 4 and figure 1). Please discuss this point.

We are not sure why the reviewer would consider this clinically irrelevant. We feel that the association only with the highest HEAT scores indicates the high degree of global dietary compliance necessary to achieve the recommended CHILD-2 dietary intake targets, which sets targets for what HEAT score might be necessary in clinical practice.

Overall, it seems to me unjustified to conclude that "... These results show promise that the HEAT may be a helpful tool to assess and track broader dietary goals in children, particularly identifying children with suboptimal dietary patterns that may put them at risk of obesity and dyslipidemia and who might benefit from targeted dietary counselling...".

We respectfully disagree. We have carefully worded this statement to imply that our study is not definitive, and that further study is necessary.

Reviewer 2 Report

This study used data of 70 enrolled participants to analyze the association of Healthy Eating Assessment Tool(HEAT) scores with components of Cardiovascular Health Integrated Lifestyle Diet (CHILD-2) or markers of adiposity or lipid biomarkers. A higher HEAT score was found to correlate with CHILD-2 total fat percent of total daily calories and dietary fiber intake, the HEAT score is also associate with BMI z-score and waist-height ratio, but there is no association between HEAT score and Lipid variables. Some concerns are listed:

1.     It is necessary to have a comprehensive comparison for HEAT and CHILD-2 at the same time analyzing the same population to show the advantages and disadvantages for the two assessment tools. CHILD-2 is a general assessment tool, when study the association of a method HEAT with another method CHILD-2, will this increase the inaccuracy?

2.     Lipid variables are more physiological and biochemical parameters indicating the disease situation, however, there is no lipid variable was significantly associated with HEAT score. Please clarify the reasons. Whether this also shows the limitation and inaccuracy of the HEAT score. Please discuss more about this.

3.     In figure 1, why the sum of poor, fair, good, excellent groups is not 1 for the proportion meeting recommendation for each standard? Please clarify.

4.     The figure quality and resolution should be improved.

5.     The association analysis without significance should also be included in the supplementary to support the description in text as proof.

6.     The appendix A and B are not available for reviewing. Since the assessment or scoring method is very important to avoid subjectivity. It is highly suggested to incorporate concise information about how to score in the method part.

7.     The content in Table 1 and 2 is totally same as the information delivered in text, please revise to avoid repetition.

8.     Line 75“and informed consent was”“,”is suggested to add.

9.     Line 76, extra blank or space before “The Hospital ”.

10.  line 155,  “percent of daily monounsaturated and polyunsaturated fat intakes”.

Author Response

This study used data of 70 enrolled participants to analyze the association of Healthy Eating Assessment Tool(HEAT) scores with components of Cardiovascular Health Integrated Lifestyle Diet (CHILD-2) or markers of adiposity or lipid biomarkers. A higher HEAT score was found to correlate with CHILD-2 total fat percent of total daily calories and dietary fiber intake, the HEAT score is also associate with BMI z-score and waist-height ratio, but there is no association between HEAT score and Lipid variables. Some concerns are listed:

  1. It is necessary to have a comprehensive comparison for HEAT and CHILD-2 at the same time analyzing the same population to show the advantages and disadvantages for the two assessment tools. CHILD-2 is a general assessment tool, when study the association of a method HEAT with another method CHILD-2, will this increase the inaccuracy?

The HEAT is a simplified dietary assessment tool. The CHILD-2 is not an assessment tool, but the recommended dietary fat and cholesterol targets for pediatric patients for LDL-C reduction. Achievement of these targets was assessed from analysis of food recall records. The same population is necessary to determine if HEAT scores are reflective of achievement of CHILD-2 dietary targets. We are not showing the advantages and disadvantages of two “tools”. We are showing how the HEAT score relates to achievement of CHILD-2 targets, as well as to markers of adiposity and lipid values.

  1. Lipid variables are more physiological and biochemical parameters indicating the disease situation, however, there is no lipid variable was significantly associated with HEAT score. Please clarify the reasons. Whether this also shows the limitation and inaccuracy of the HEAT score. Please discuss more about this.

The reasons why there was no association of HEAT with the lipid variables is because of the heterogeneity of the underlying lipid conditions in our clinic, which would be more likely to drive the lipid levels than dietary compliance. This is not reflective of any limitation or inaccuracy of the HEAT assessment. Since this study is cross-sectional in nature, a further study would be needed to define the association of changes in HEAT score with changes in lipid variables, adjusting for changes in medication, age, adiposity, and other important covariates.

  1. In figure 1, why the sum of poor, fair, good, excellent groups is not 1 for the proportion meeting recommendation for each standard? Please clarify.

The figure and the figure legend have been redone to clarify that the height of the bars reflects the percentage of patients within each HEAT score category that met that particular CHILD-2 target. Hence, the percents should not add up as the reviewer suggests.

  1. The figure quality and resolution should be improved.

The figures have been redone and improved.

  1. The association analysis without significance should also be included in the supplementary to support the description in text as proof.

All of the association analyses performed are provided in the body of the paper with the exception of the general linear models of the HEAT score with the lipid variables. This is now provided in a Supplemental Table.

  1. The appendix A and B are not available for reviewing. Since the assessment or scoring method is very important to avoid subjectivity. It is highly suggested to incorporate concise information about how to score in the method part.

We have replaced the Table with the HEAT score components with a Figure that shows the actual instrument. We have added more detail regarding scoring in the methods.

  1. The content in Table 1 and 2 is totally same as the information delivered in text, please revise to avoid repetition.

We have eliminated this repetition in the text.

  1. Line 75,“and informed,consent was”,“,”is suggested to add.

Informed consent is the correct phrase, adding a comma alters the meaning.

  1. Line 76, extra blank or space before “The Hospital ”.

This has been corrected.

  1. line 155,  “percent of daily monounsaturated and polyunsaturated fat intakes”.

This has been corrected.

Round 2

Reviewer 1 Report

1. Lines 194-196: "In this prospective cross-sectional study of pediatric patients with dyslipidemia, we noted that the  HEAT score was modestly associated with CHILD-2 daily fat percentage targets and more consistently associated with the markers of adiposity, specifically BMI z-score and waist:height ratio"

     As I noticed in my previous review, a score which provides just some (and limited) hints on markers of adiposity is sparsely worth the labour and time needed for its calculation. Much easier to weigh the patient and measure height (by the way, I apologize for confusing height with girth but that does'nt change much)

 2.  Lines 197-199: "These results show promise that the HEAT may be a helpful tool to assess and track broader dietary goals in children, particularly identifying children with suboptimal dietary patterns that may put them at risk of obesity and dyslipidemia and who might benefit from targeted dietary counselling"
In my opinion, the data do not show promise but rather cannot provide a definite answer to the important point under investigation and need therefore further evaluation.

     Please rephrase the conclusion which the Authors themselves seem to share in lines 249-250: "These results suggest that the HEAT tool may have limited utility in the assessment of compliance and meeting the specific targets of the CHILD-2 diet."

3.   Line 142-146: " The underlying dyslipidemia was heterozygous familial hypercholesterolemia for 33 (47%) patients (19 treated with a statin), combined dyslipidemia for 16 (23%) patients (2 treated with medication), mild dyslipidemia with elevated LDL-C for 14 (20%) patients, isolated low HDL-C for 4 (13%) patients, and elevated lipoprotein(a) for 3 (10%) patients"
Please clarify:
-Why 60% of children and adolescent with heterozygous familial hypercholesterolemia were not on drug tretment
-Mild dyslipidemia: please quantify

     -What the evidence about dietary treatment of isolated low HDL (how much low, please quantify) in the light of the controversy about the prognostic and clinical value of HDL values in cardiovascular disease

-How high was lipoprotein (a) and what the evidence about treatment whatsoever of this abnormality of still uncertain clinical significance

Author Response

Reviewer 1:

  1. Lines194-196: "In thisprospective cross-sectional study of pediatric patients with dyslipidemia, we noted that the  HEAT score was modestly associated with CHILD-2 daily fat percentage targets and more consistently associated with the markers of adiposity, specifically BMI z-score and waist:height ratio"

     As I noticed in my previous review, a score which provides just some (and limited) hints on markers of adiposity is sparsely worth the labour and time needed for its calculation. Much easier to weigh the patient and measure height (by the way, I apologize for confusing height with girth but that does'nt change much)

This is not the point. The HEAT is not a marker of adiposity, and we don’t state that it is. BUT, poor dietary habits and composition are associated with an increased risk of obesity and associated combined dyslipidemia. The association of poor HEAT score and increased markers of adiposity is relevant. In addition, use of the HEAT tool highlights dietary areas for more specific intervention. We standby the statement as we have written it.

  1. Lines 197-199: "These results show promise that the HEAT may be a helpful tool to assess and track broader dietary goals in children, particularly identifying children with suboptimal dietary patterns that may put them at risk of obesity and dyslipidemia and who might benefit from targeted dietary counselling"
    In my opinion, the data do not show promise but rather cannot provide a definite answer to the important point under investigation and need therefore further evaluation.

We have changed the sentence to: These results may suggest that the HEAT may be a more pragmatic tool to assess and track broader dietary goals in children, particularly identifying children with suboptimal dietary patterns that may put them at risk of obesity and dyslipidemia and who might benefit from targeted dietary counselling.

     Please rephrase the conclusion which the Authors themselves seem to share in lines 249-250: "These results suggest that the HEAT tool may have limited utility in the assessment of compliance and meeting the specific targets of the CHILD-2 diet."

We have rephrased the conclusions to: Our results suggest that the HEAT tool may have limited utility in the assessment of compliance and meeting the specific targets of the CHILD-2 diet. However, the HEAT tool may be a more pragmatic and simplified way of assessing and tracking broader dietary goals in clinical practice.

  1. Line 142-146: " The underlying dyslipidemia was heterozygous familial hypercholesterolemia for 33 (47%) patients (19 treated with a statin), combined dyslipidemia for 16 (23%) patients (2 treated with medication), mild dyslipidemia with elevated LDL-C for 14 (20%) patients, isolated low HDL-C for 4 (13%) patients, and elevated lipoprotein(a) for 3 (10%) patients"
    Please clarify:
    -Why 60% of children and adolescent with heterozygous familial hypercholesterolemia were not on drug tretment

All of the patients who had FH but were not on a statin were either too young at the time of assessment or had LDL-C levels just below our threshold for starting a statin (LDL-C 4.1 mmol/L).

-Mild dyslipidemia: please quantify

Patients with mild dyslipidemia had elevated LDL-C levels below 3.5 mmol/L without a family history suggestive of familial hypercholesterolemia. A number of these patients also have low HDL-C and/or high triglycerides, but not suggestive of a combined dyslipidemia.

-What the evidence about dietary treatment of isolated low HDL (how much low, please quantify) in the light of the controversy about the prognostic and clinical value of HDL values in cardiovascular disease.

All patients with isolated low HDL-C had levels <1.00 mmol/L. Pediatric lipid guidelines recommend that patients with isolated low HDL-C optimize management of other risk factors and compliance with healthy lifestyle recommendations, including achievement of dietary targets. Of note, 3 of the 4 patients with isolated low HDL-C were overweight, further indicating the need to achieve the broader dietary targets indicated by the HEAT assessment tool.

-How high was lipoprotein (a) and what the evidence about treatment whatsoever of this abnormality of still uncertain clinical significance.

Similar to isolated low HDL-C, the goal of healthy lifestyle management in the setting of elevated lipoprotein(a) is to optimize management of other risk factors. The patients in our clinic who we see for lipoprotein(a) are patients who have had neonatal stroke and a thrombophilia work-up. Patients with levels above 50 mg/dL are referred and followed in our lipid clinic. The 3 patients with elevated lipoprotein(a) all had level above 50.

Reviewer 2 Report

There are concerns about the novelty and significance for the study.

Author Response

No comments from Reviewer 2.